# A Promising and Challenging Approach: Radiologists’ Perspective on Deep Learning and Artificial Intelligence for Fighting COVID-19

**DOI:** 10.3390/diagnostics11101924

**Published:** 2021-10-18

**Authors:** Tianming Wang, Zhu Chen, Quanliang Shang, Cong Ma, Xiangyu Chen, Enhua Xiao

**Affiliations:** 1Department of Radiology, The Second Xiangya Hospital, Central South University, Changsha 410011, China; timwong73@csu.edu.cn (T.W.); chenzhu415@csu.edu.cn (Z.C.); sqljsdh@csu.edu.cn (Q.S.); mcismc@126.com (C.M.); chenxiangyu@csu.edu.cn (X.C.); 2Department of Radiology, Xiangya Hospital, Central South University, Changsha 410008, China; 3Molecular Imaging Research Center, Central South University, Changsha 410008, China

**Keywords:** machine learning, deep learning, artificial intelligence, medical imaging, COVID-19

## Abstract

Chest X-rays (CXR) and computed tomography (CT) are the main medical imaging modalities used against the increased worldwide spread of the 2019 coronavirus disease (COVID-19) epidemic. Machine learning (ML) and artificial intelligence (AI) technology, based on medical imaging fully extracting and utilizing the hidden information in massive medical imaging data, have been used in COVID-19 research of disease diagnosis and classification, treatment decision-making, efficacy evaluation, and prognosis prediction. This review article describes the extensive research of medical image-based ML and AI methods in preventing and controlling COVID-19, and summarizes their characteristics, differences, and significance in terms of application direction, image collection, and algorithm improvement, from the perspective of radiologists. The limitations and challenges faced by these systems and technologies, such as generalization and robustness, are discussed to indicate future research directions.

## 1. Introduction

The rapid spread of the COVID-19 worldwide developed into a global pandemic, leading to a large-scale medical system burden and economic crisis. The detection and diagnosis of COVID-19 rely on the positive results of the reverse transcription-polymerase chain reaction (RT-PCR); however, false-negative results may occur due to the influence of various factors such as low viral load [1]. Chest imaging examinations, including CT and CXR images, are currently among the most important tools for screening and diagnosing COVID-19 patients [2]. However, the generation of a large amount of image data increases the workload of radiologists. Moreover, the massive information from radiologic images is also difficult to observe and summarize by sight. In recent years, due to the availability of high-performance computers and the large amount of data generated in the information age, the application of machine learning and artificial intelligence in many fields has made exciting new progress, which has also stimulated innovation in the development of artificial intelligence technology based on medical imaging [3] (Figure 1). The widespread application of ML and AI has led to massive information in medical imaging, playing a pivotal role in the fight against the COVID-19 epidemic [4] (Figure 2). However, generalization and robustness are still the main challenges of DL and AI models [5]. Sample size of datasets, nonstandard protocols of imaging, image segmentation, availability of public datasets, multimodal models combining medical imaging and clinical information, and extensive clinical validation and applications remain the main limitations.

Herein, the application, existing problems, and possible solution of machine learning and artificial intelligence systems based on medical images are introduced, such as diagnosis and classification, treatment decision-making, curative effect evaluation, prognosis prediction, and other relevant aspects for patients affected by COVID-19. By conducting a rapid review of published studies and the existing experience of DL and AI in COVID-19 radiological examinations, the aim of this article is to evaluate the feasibility and the effectiveness of available COVID-19 diagnostic tools, which may be helpful to develop better intelligent models by medical experts and technical researchers.

## 2. Methods

A combination of three group of keywords including disease(“COVID 19”, “SARS-CoV-2”, “NPC”), imaging examinations(“CT”, and “X-ray”), together with algorithm technology(“machine learning”, “deep learning”, and “artificial intelligence”) was used to search for articles in four databases: PubMed, Web of Scicence, Elsevier/SD, and IEEE; all types of research were evaluated, including case reports, original research, clinical guidelines, and systematic reviews. Through reading abstracts and reviewing the references of included studies, the published studies were finally screened and classified. We set the criteria for inclusion and classification. Only research written in English was included; preprints that were not peer-reviewed were excluded. The application and effectiveness of these systems were discussed from the perspective of radiologists. Although the types and development of algorithms were not the focus of this review, they were not excluded Figure 3.

## 3. Application of Detection, Diagnosis, and Classification of COVID-19

Chest X-ray and CT are widely used to diagnose and screen COVID-19 patients from suspected patients and make differential diagnosis from patients with other lung diseases [6,7,8]. Quantitative analysis of suspicious COVID-19 lesions on CT images is a basic application of DL and AI models. Zhang et al. [9] used the uAI intelligent auxiliary analysis system to accurately evaluate CT images. The system generates a Hounsfield unit (HU) histogram in the infected area to achieve precise quantification and visualization of the lesion. Their generalized linear mixed model showed that the favored site for COVID-19 pneumonia was the dorsal segment of the right lower lobe. Li et al. [10] developed a fully automated framework DL model named COVID-19 Detection Neural Network (COVNet) and analyzed chest CT images to distinguish COVID-19, community-acquired pneumonia (CAP), and other chest diseases. When detecting COVID-19 and CAP, the area under the curve (AUC) of the receiver operating characteristic (ROC) was 0.96 and 0.95, respectively. The study of Barbosa et al. achieved similar results [11]. This research shows that the DL and AI models trained by collecting large numbers of patient image datasets can efficiently screen and diagnose suspected COVID-19 patients. Jin et al. [12] reported that their multiclass AI diagnosis model based on deep convolutional neural networks (CNN) achieved an AUC of 0.9781 in the test cohort of 3199 CT scans to diagnosing COVID-19, CAP, and influenza (Type A/B) and nonpneumonia lesions, and for the test in the two publicly available open-source datasets CC-CCII and MosMedData, it was 0.9299 and 0.9325, respectively. The results also indicate that the diagnostic ability of the trained AI system may be affected in external tests, and the application of AI requires learning with a larger range of data and larger samples to improve repeatability. Harmon et al. [13] adapted a DL model on CT data from a group of multinational patient cohorts to identify COVID-19, which showed an accuracy and sensitivity of 90.8% and 84%, respectively, thus suggesting that the trained AI system derived from heterogeneous multicountry training data provides acceptable performance indicators for chest CT classification of COVID-19 infection. Zhang et al. [14] developed a CT-based AI diagnosis system by training and verifying the CT data of 3777 patients. Their AI approach could differentiate COVID-19 from two other classes with 92.49% accuracy, 94.93% sensitivity, 91.13% specificity, and an AUC of 0.9797, when validated on their internal dataset. The system has also shown similar diagnosis performance on three prospective test sets, open-source CT datasets, and datasets outside of China, proving the general applicability of the AI diagnostic approach.

Although CT was recommended in the diagnosis of COVID-19 at different stages of the disease in mainland China, the British Society of Thoracic Imaging and the American College of Radiology currently recommend CXR as the first line of COVID-19 imaging [15,16]. As a result, a large amount of CXR image data is produced because X-ray images are easier to obtain and more numerous than CT images; furthermore, AI models using X-ray datasets have been extensively developed. Wang et al. [17] developed a deep learning system trained on a heterogeneous multicenter dataset containing 145,202 images to distinguish viral pneumonia from other types of pneumonia and absence of pneumonia, obtaining sensitivity, specificity, and an AUC of 87.04%, 92.94%, and 0.968, respectively. Wehbe et al. and Keidar et al. implemented comparable machine learning approaches for detection, achieving an AUC of 0.90 and 0.96, respectively [18,19].

## 4. Application of Early Intervention and Decision-Making

DL and AI systems can extract massive amounts of valuable clinical decision information by analyzing medical images and clinical data from COVID-19 patients, thus revealing the great potential for the use of chest X-rays and CT for quantifying the lesions and assessing progression and related medical indicators. Zhang et al. [14] used quantitative measurements generated by AI to evaluate the effect of drug treatment on changes in lesion size and volume in CT scans, reporting the potential of the AI system to objectively and quantitatively evaluate the effects of drug treatment, which can assist medical staff in making management decisions for COVID-19 patients. Cai et al. [20] built random forest (RF) models for classification and regression to evaluate the disease severity (moderate, severe, and critical) and to predict by CT quantification the need and length of ICU hospital stay, the duration of oxygen inhalation, hospitalization, sputum NAT positive, and patient prognosis. The results showed that the root-mean-square error of the RF regressor to predict the length of hospital stay was 0.88 weeks, oxygen inhalation time was 0.92 weeks, and the AUC predicting ICU treatment and prognosis (partial recovery vs. prolonged recovery) were 0.945 and 0.960, respectively. The indicators and parameters of hospitalized critically ill patients include ICU treatment decisions, ICU duration, oxygen inhalation, and hospitalization, which are critical to clinical management. However, affected by the small patient sample size and the binary classification problems, the model of ICU demand has slightly worse results, and the performance of the model for predicting prognosis dropped precipitously, suggesting that the multitask model is affected by ML algorithms and data types; thus, it is difficult to maintain excellent performance in every task. Wang et al. [21] trained and verified a deep learning system that successfully stratified patients into high-risk and low-risk groups with significant differences in hospital stay time. Deep learning provides conveniently available tools for quickly identifying potentially high-risk patients, which may help optimize medical resources and provide early prevention before patients develop severe symptoms. Furthermore, Yue and colleagues [22] generated a CT radiomics models using logistic regression (LR) and random forest (RF), and effectively discriminated short-term and long-term hospital stay among patients with COVID-19; the AUC of the CT radiomics models, based on six second-order features, were 0.97 (LR) and 0.92 (RF), respectively. These studies have concluded that DL and AI are beneficial decision-makers that can provide more medical information, optimize medical resources, and help identify necessary steps for prevention before patients manifest severe symptoms.

## 5. Comparison of Medical Images with Clinical Data

A large number of clinical parameters and imaging data, which are generated during the diagnosis and treatment of COVID-19 patients, can be compared and correlated through DL and AI systems. Zhang et al. [14] selected three lesion indicators, total lesions (named lesions), ratio of lymphocytes (CL), and ground-glass opacity (GGO), for the entire lung field for correlation analysis, finding that C-reactive protein (CRP), age, serum lactate dehydrogenase (LDH), maximum body temperature (Tmax), and neutrophil/lymphocyte ratio had a strong positive correlation with lesion characteristics. This suggested that the AI system identified important clinical markers correlated with the COVID-19 lesion properties. Wang and his team [23] used a DL system to perform quantitative segmentation and calculation of the lesion area in the high-resolution CT image of COVID-19 patients. Comparison of the differences of the percentage of infection (POI) of whole lung and relationship between POI and clinical laboratory examination in all patients showed that the POI of the whole lung was negatively correlated with percentage of peripheral blood lymphocytes (L%) and lymphocytes (LY) count, and positively correlated with the percentage of neutrophils (N%). This indicated that both parameters could be used as prognostic indicators, having warning significance and contributing to clinical intervention for patients. Children are not completely immune to COVID-19. In order to determine whether CT is necessary for children as a vulnerable group, Ma et al. [24] tried to determine whether children required CT as a vulnerable group. They created a machine learning model based on advanced decision trees to predict the CT results of children with COVID-19 by combining clinical symptoms and laboratory data, which showed that age, lymphocytes, neutrophils, ferritin, and C-reactive protein were the most relevant clinical indicators for predicting CT results among pediatric patients with positive RT-PCR detection. The decision support system could effectively predict CT results. Therefore, their findings indicated that the use and necessity of CT for pediatric patients should be reconsidered, as the model can effectively predict CT outcomes. Fung et al. [25] proposed a self-supervised two-stage deep-learning model named SSInfNet to segment COVID-19 lesions (including ground-glass opacity and consolidation) from chest CT images. The authors identified several CT image phenotypes that mediate the potential causal relationship between underlying diseases and age with COVID-19 severity.

## 6. Comparison of the Diagnostic Performance of DL and AI Systems with Radiologists

The effectiveness of DL and AI systems in diagnosing COVID-19 is a basic condition for clinical application and is often compared with the diagnostic performance of radiologists. Murphy et al. [26] used data from 24,678 chest X-rays for training and the other 1540 images for post-training verification to develop an AI system (CAD4COVID-XRay). The ROC analysis of the independent diagnosis performance of the six readers and the AI system showed that the AI system could correctly classify CXR images as COVID-19 pneumonia with 0.81 AUC. The system also significantly outperformed each reader at their highest possible sensitivity. It is suggested that the diagnostic performance of trained DL and AI systems may be a useful tool for radiologists, or may be used as an auxiliary tool for medical teams without radiology expertise. Wehbe et al. [18] developed an AI diagnostic system that used DeepCOVID-XR and compared the AI algorithm with the diagnostic performance of five experienced thoracic radiologists. ROC-AUC of DeepCOVID-XR was 0.88 compared with the consensus AUC of 0.85. With consensus interpretation as the reference standard, the AUC of DeepCOVID-XR was 0.95. The authors believed that DeepCOVID-XR is more reliable than the study by Murphy et al. for predictions produced by DeepCOVID-XR are in line with the radiological diagnosis by a consensus of experts. Hwang et al. [27] observed that DL-based CAD may improve the performance of readers, while the performance of nonradiologists significantly improved in the CAD-assisted interpretation. Furthermore, inter-reader agreement among physicians showed significant improvement when assisted with the CAD. To distinguish COVID-19 from other pneumonia, Bai et al. [28] used the EfficientNet B4 deep neural network architecture trained on data from CT scans of 1186 patients. Their results showed that the radiologists achieved a higher average accuracy (90% vs. 85%), sensitivity (88% vs. 79%), and specificity (91% vs. 88%) with the assistance of AI models. It is suggested that radiologists with the assistance of AI systems could improve the accuracy of diagnosing COVID-19. Moreover, Wang et al. [17] used a fully automated DL pipeline for the quantification of the severity scores in CXR images to evaluate the correlation between the DL model and radiologists. The results obtained showed the Pearson correlation coefficient was 0.81, and the AI system could predict the severity of COVID-19 pneumonia with an AUC of 0.868, indicating that the severity index of radiologists and the AI system had a strong linear relationship. Furthermore, the DL system showed that its diagnostic performance was comparable to that of senior radiologists for the CXR test set, while it also improved the performance of junior radiologists. 

It can be seen from these studies that the detection and diagnosis performance of AI models based on COVID-19 CXR or CT images is similar to that of experienced radiologists and may help improve the reliability of physicians’ diagnoses, especially for junior radiologists and nonradiologists.

## 7. Prediction of Curative Effect Evaluation and Prognosis in COVID-19

Objectively quantifying the degree of disease by the percentage of COVID-19 lesions in the lung parenchyma is currently the most important application of AI in the evaluation of the COVID-19 patient, which can contribute to evaluating and monitoring the progression of the disease. In order to further analyze the clinical and radiological characteristics that lead to critical illness, it is necessary to develop artificial-intelligence-assisted models to estimate clinical prognosis. Shan et al. [29] used the VB-Net neural network DL to segment the COVID-19 infection area on the CT scan and compared three indicators, the Dice similarity coefficient, volume difference, and percentage of infection (POI), between automatic and manual segmentation. This DL-based segmentation system produced a Dice similarity coefficient of 91.6% ± 10.0% between automatic and manual segmentation when using multiple lung lobes and bronchopulmonary segment infections. Additionally, when quality (MOI) was used as a feature for severity prediction, the best accuracy of severity prediction was 73.4% ± 1.3%, indicating that the quantitative techniques of the DL system have potential clinical application value in predicting the severity of COVID-19. Li et al. [30] developed an algorithm based on a convolutional Siamese neural network to measure the severity of COVID-19 on CXR automatically. Their results showed that the pulmonary X-ray severity (PXS) score was correlated with the radiological chest disease severity score of COVID-19 on CXR in the internal and external test sets, and the ROC–AUC of PXS score predicting intubation or death within three days after admission was 0.80 (95% CI 0.75–0.85). A similar method is well established and used in other studies such as that reported. Lessmann et al. [31] adapted a deep learning algorithm system called CORADS-AI and followed a standardized reporting scheme to assign CO-RADS scores for suspected COVID-19 and identify COVID-19 by chest CT examinations, where the AUC of the internal cohort and the external cohort were 0.95 and 0.88, respectively, and the CT severity score was assigned to the extent of parenchymal involvement of each lobe, which was consistent with the readings of eight independent human observers. This proposed method verified the accuracy of the AI system and compared it with human observers, which avoids nonstandardized and uncalibrated output and enhances the interpretability of the output. The DL system developed by Fang et al. effectively combined the dynamic sequence of chest CT scans and static clinical information from 1040 COVID-19 patients with mild symptoms at different times after hospitalization. Their results showed that the system could effectively identify valuable indicators to predict the malignant progression of COVID-19, with an average AUC of 0.920 in the single-center study and average AUC of 0.874 in the multicenter study [32]. Zhang’s team conducted research based on the quantitative chest lesion features and clinical parameters extracted by the AI system. Their results showed that the high-risk group (C-score ≥ 0.5) had a much lower survival probability and a lower risk compared to the low-risk group (C-score < 0.5), indicating that the combination of chest imaging data and clinical data could significantly promote prognostic evaluation [14].

Although numerous studies report the efficacy evaluation and prognosis prediction of COVID-19 based on imaging, there are relatively few studies on imaging risk factors for poor prognosis of COVID-19 [33,34]. Research by Bartolucci et al. analyzed the volume and texture of the lesion area in the CT of 115 COVID-19 patients; results suggested that the mixed radiology model, including the percentage of pulmonary consolidation, performed significantly better (AUC = 0.82) compared to the blood-laboratory–arterial-gas analysis feature alone (AUC = 0.71) in the validation set predicting ICU admission [33]. This suggested a mild but significant incremental value in predicting ICU occupancy by evaluating the volume of the consolidated lung in CT of COVID-19 patients. Yu et al. [34] used a dedicated multitask DL algorithm to build an AI system to quantitatively analyze the volume, density, location, GGO, and substance of the patient’s lesions, after which they used multivariable logistic regression to evaluate the imaging features and risk factors related to prognostic endpoints (admission to ICU, acute respiratory failure occurrence, or shock during hospitalization). Their results showed that older age at admission and a larger area of consolidation in the upper lung was more closely related to the poor prognosis of COVID-19 patients. In addition, Liang et al. [35] developed a deep learning-based survival model for predicting the risk of critical illness in COVID-19 patients based on clinical characteristics (X-ray abnormalities included) at the time of admission. Their results showed that internal verification performance (0.894) and external verification of three independent cohorts (0.890, 0.852, and 0.967) achieved good consistency indexes. This suggested that the AI model could be used to classify patients at admission and identify patients at risk of serious disease.

DL and AI systems can be used to monitor and quantify disease progression and understand the time evolution characteristics of COVID-19 lesions, reducing the subjectivity of radiologists in comparing chest CT images before and after treatment. They are also conducive to accurate treatment. 

## 8. Follow-Up and Recovery Evaluation after Discharge

Although there are currently many studies focusing on the evaluation of the radiology and clinical physiological outcomes of recovered COVID-19 patients [31,36], there are few studies using DL and AI models of radiological images for post-discharge follow-up and recovery evaluation. Meng et al. [37] applied a novel DL technique by learning the chest CT scan features of 270 hospitalized COVID-19 patients with two consecutive negative RT-PCR tests (sampling interval > 1 day). The discharge standard set by DL in the study is that the total lesions are less than 50% of the total lung volume. The results showed there are no positive test results or progression of pneumonia among the 230 discharged cases, thus achieving the aim to standardize the discharge criteria at a “square cabin” hospital; furthermore, the study found that vascular dilatation has the potential to predict the short-term prognosis of COVID-19 patients. Hu et al. [38] calculated the degree of pulmonary fibrosis with the volume of fibrosis on the chest CT image using an AI assistant program machine. The basic inflammatory molecular levels at admission and after discharge were compared. The results showed that plasma interferon-γ(IFN-γ) levels were twofold lower than those without fibrosis, and there was a negative correlation between pulmonary fibrosis and basal circulating IFN-γ levels in the recovery phase. It shows that the decrease of IFN-γin circulation is a risk factor for pulmonary fibrosis of COVID-19.

## 9. Constructing Image Multimodal Models to Evaluate COVID-19

Many biological data have been accumulated with the spread of the COVID-19 epidemic and numerous infected patients generating epidemic contact history, clinical data, laboratory examinations, and imaging data. Researchers have constructed various multimodal DL and AI systems from different angles to fully mine the information contained therein, and then transform the data into reliable and referable information in the diagnosis and treatment of COVID-19 [39]. Li et al. [40] reported two false-negative results of rRT-PCR for SARS-CoV-2 infection and mentioned that a comprehensive method combining DL, CT features, and rRT-PCR results might be used to ensure early screening and accurate diagnosis of COVID-19 in clinical practice. Mei et al. [41] used AI algorithms by integrating chest CTs with clinical symptoms, exposure history, and laboratory tests to quickly diagnose COVID-19 patients, revealing that the joint model outperformed the model trained on CT images only and the model trained on clinical information only. Furthermore, the AI model also correctly identified several COVID-19 patients with a normal CT scan who were misdiagnosed by radiologists (17/25) and confirmed by rRT-PCR. Purkayastha et al. [42] collected information from 981 patients with COVID-19 from a multi-institutional international cohort to extract radiomics features of chest CTs and clinical information; they trained a machine learning system to predict severity and time-to-event model to predict progression to critical illness in combination with CT imaging omics and clinical variables. The model successfully predicted the risk of progression on the third, fifth, and seventh day, which suggested that a multimodal combination of CT radiomics characteristics and clinical variables could predict the severity and progression of COVID-19 to critical illness with high accuracy. Moreover, Schaffino et al. [43] used machine learning models to assess lung parenchyma and vascular damage. Their results showed that the best models of SVM and MLP considered the same ten input features, yielding an AUC of 0.747 and 0.844, respectively. Furthermore, in this model that integrated clinical and radiological data, pulmonary artery diameter was the third-largest predictor after age and physical involvement, which helped to reliably predict mortality during hospitalization, underscoring the value of vascular indicators for improving severity stratification patients. Jiao et al. [44] examined an EfficientNet deep neural network based on CXR images to estimate the disease severity and progression. When deep learning features were added to DL system of clinical data which used to predict progress, the concordance index (C-index) of both internal and external tests increased. The prognostic performance of the image and clinical data combination model was significantly better than the severity score and clinical data combination on the internal and external tests. It can be concluded that the performance of the multimodal model fused with imaging and clinical data is significantly better than that of models based solely on the image or clinical data.

## 10. Development of New DL and AI Algorithms

Due to the lack of standard requirements for datasets, there are significant differences in the dataset sources used by different studies and the image scanning conditions of each medical center or database. Researchers studied the performance of various algorithms under different data sources. Shiri et al. [45] used ultralow-dose and full-dose CT images as input/output for training, testing, and external validation sets to implement full-dose prediction technology. It was found that the radiation dose expressed by CT Dose Index (CTDIvol) was reduced by 89%, and the DL algorithm could predict standard full-dose CT images with acceptable quality for clinical diagnosis of COVID-19 patients, while radiation dose could be effectively reduced, the deep learning algorithm failed to restore the correct lesion structure density for some patients who were regarded as outliers. Therefore, further research and development are necessary to solve these limitations. Sengupta et al. [46] developed a new quantum neural network (QNN) algorithm, where the system running time observed on quantum-optimized hardware was 52 min, while on K80 GPU hardware, the observed model running time was 1 h and 30 min for similar sample sizes. Simulation experiments showed that the performance of QNN in accuracy measurement was better than DNN, CNN, and 2D CNN by more than 2.92%, while the average recall rate was about 97.7%. Rapid clinical prognostic analysis could be achieved by using CT images. Wu et al. [47] proposed a weakly supervised deep active learning framework called COVID-AL to diagnose COVID-19. A hybrid sampling strategy was used to select samples with high uncertainty and diversity in each round of active learning, which greatly reduced the cost of training models for manually labeled chest CT scans. Wang et al. [48] developed a deep learning system and radiomics models for predicting and classifying COVID-19 and non-COVID-19 viral pneumonia. Their results indicated that both models achieved > 73% sensitivity and >75% specificity in the external validation cohort, while the performance of the radiomics lasso classifier was slightly superior. The diagnostic performance of human experts improved when the joint deep learning–radiomics model was used. Transfer learning and data enhancement were common strategies to combat data scarcity in deep learning algorithms among the existing literature, and it retrains the deep model on a large-scale dataset and fine-tunes it on the target image set of COVID-19 [49,50]. In order to further evaluate the discriminative ability of different models, ensemble learning is deployed, and multiple deep networks are used to evaluate the final results. Fouladi et al. [51] proposed ResNet-50, VGG-16, Convolutional Neural Network (CNN), Convolutional Autoencoder Neural Network (CAENN), and Machine Learning (ML) methods to classify and compare chest CT images of COVID-19 patients; nearest neighbor (NN) had the highest performance with an accuracy of 94%. The limitation of this study was that it did not use image preprocessing and data enhancement techniques, and the use of pre-trained networks or data enhancement may improve accuracy. Since direct transmission between datasets from different fields may lead to poor performance, researchers have developed various strategies to mitigate the impact of domain differences on transmission performance [52].

## 11. Other Nonimaging-Related Applications of DL and AI

The most common use of deep learning and artificial intelligence based on medical images is for diagnosing and evaluating COVID-19 patients. In addition to using medical image data (Table 1), DL and AI are also widely used in tasks related to patient outcomes, such as the diagnosis, assessment of severity, prediction of death risk, and length of hospital stay of COVID-19 patients. Imran et al. [53] developed an AI4COVID-19 system to diagnose COVID-19 by recording three 3 s cough sounds and sending them to an AI engine running in the cloud. Their results showed that the system could distinguish between COVID-19 coughs and several non-COVID-19 coughs to screen for COVID-19. DL and AI technologies have also had a vital role in epidemic monitoring and development prediction [54,55]. Ribeiro and colleagues [56] developed an effective short-term prediction model for forecasting the number of short-term cases in Brazil. Compared with the comparative model, support vector regression (SVR) and stacked integrated learning achieved better adoption of the adopted standards. Based on a hybrid method using an autoregressive integrated moving-average model together with a wavelet-based prediction model, a real-time prediction model was proposed to predict future epidemics in different countries, which may help with the effective allocation of medical resources and serve as an early warning reference for government decision-makers [57]. In addition, it can also be used for epidemiological investigation and tracking of close contacts, discovery or reuse of drugs and vaccines, evaluation of their safety, and optimization of treatment programs [58]. The potential capabilities of DL and AI in exploring COVID-19 vaccine antibodies and therapeutic drugs are unparalleled and can accelerate the reuse and development of COVID-19 drugs [59,60,61]. Ong summarized the research on deep learning for COVID-19 and applied Vaxign and the newly developed machine-learning-based Vaxign-ML reverse vaccinology tool to predict COVID-19 vaccine candidates [62]. Researchers from the United States and South Korea jointly proposed a new molecular converter-drug target interaction model to solve the needs of antiviral drugs for the treatment of COVID-19 [63].

## 12. Existing Limitations and Challenges

After the COVID-19 outbreak, data-driven DL and AI research made a breakthrough in the field of medical imaging. These studies all claimed that these algorithms could diagnose or predict COVID-19 based on CXR and CT images. Although some medical centers or medical imaging alliances have disclosed some COVID-19 dataset resources [64] (Table 2), there is low image quality, inconsistent CT scan parameter settings, small size samples, irregular data labeling with strong subjectivity, and dataset sources that are not uniform. Due to nonstandard datasets, algorithm bias, nonrepeatability, lack of sufficient external and forward-looking verification, and difficulty in simultaneously considering CT and CXR issues, it is difficult to clinically verify DL and AI models in complex cases and actual open scenarios, thus limiting their clinical application. Generalization and robustness have become the key and urgent problems that limit the practical application of these data-driven DL and AI model systems. Due to these limitations and challenges, most of the models in the research have no practical and clear application value and have not become the final solution for the current clinical diagnosis and treatment of COVID-19, but there are some relatively practical and more explanatory resources applied by radiologists [65]. Currently, most COVID-19 imaging is made without artificial intelligence and that human sight is remains necessary [66].

With increasing research on the application of AI systems in COVID-19, there are increasingly more studies using similar methods.Partial code access of COVID-19 DL and AI models based on radiology images were listed in Appendix A. Overall, the number of high-quality studies is relatively small, and the lack of external verification or prospective verification is a common flaw. Most studies tend to collect data from multiple institutions for testing, but lack external verification of data from multiple institutions and datasets in order to improve generalization of the AI system. The performance of models using CT data is usually better than models using CXR data; the main reason was considered to be its inherent defect of CXR images. Ordinary CXR images are the sum of the effects of X-rays on all tissues between the X-ray source and the capture membrane; compared with CT images, the tissue structure in X-rays is unclear and lacks three-dimensional information. All current models or algorithms are evaluated based on their performance. The performance of some models has been improved by improving algorithms, but these improvements are not always effective. There is no algorithm that always performs better than other algorithms and is suitable for all application types and models. The diagnostic performance of the AI system of Zhang et al. in external verification is inferior to its performance in internal testing, and the model used to assess disease prognosis is also inferior to internal testing [14], thus, there is no best fir, only a better fit. When the sample size is sufficiently large, strategies such as transfer learning will affect the calculation efficiency and generalization ability of the system. More advanced models and algorithms provided by engineers and technicians are looked forward to in the future.

The construction of a highly standardized database is essential to improve the accuracy and generalization of the model. In addition, the development of DL and AI algorithms greatly depend on the close communication between engineer, radiologist, and clinicians, and strengthened multidisciplinary communication and cross-complementation. The establishment of multilabel and/or hierarchical classification technology evaluation and benchmark solutions will greatly promote the prevention and control of infectious diseases.

## 13. Conclusions

Since the outbreak of COVID-19, numerous researchers worldwide have extensively studied how to quickly and correctly diagnose COVID-19 and evaluate and control the condition. Machine learning and artificial intelligence with high reliability and huge potential in various applications were adopted by numerous healthcare providers. From multiple angles, this article introduced studies that explored DL and AI technologies based on medical imaging data so as to solve the troubles and challenges encountered in reality and directly point out the limitations of current studies. Overcoming these challenges may lead to significant progress in the use of artificial intelligence technology to combat COVID-19 and future epidemics.

## Figures and Tables

**Figure 1 diagnostics-11-01924-f001:**
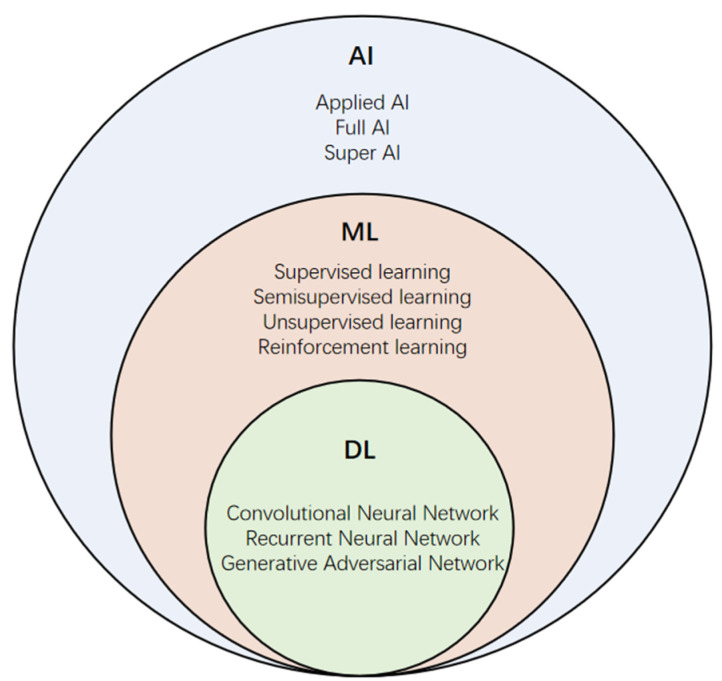
The relationship and main types of AI, ML, and DL.

**Figure 2 diagnostics-11-01924-f002:**
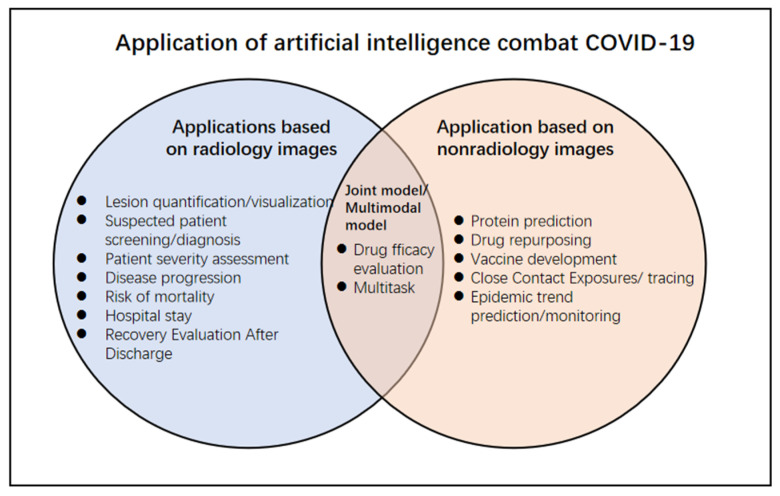
Application of artificial intelligence to combat COVID-19.

**Figure 3 diagnostics-11-01924-f003:**
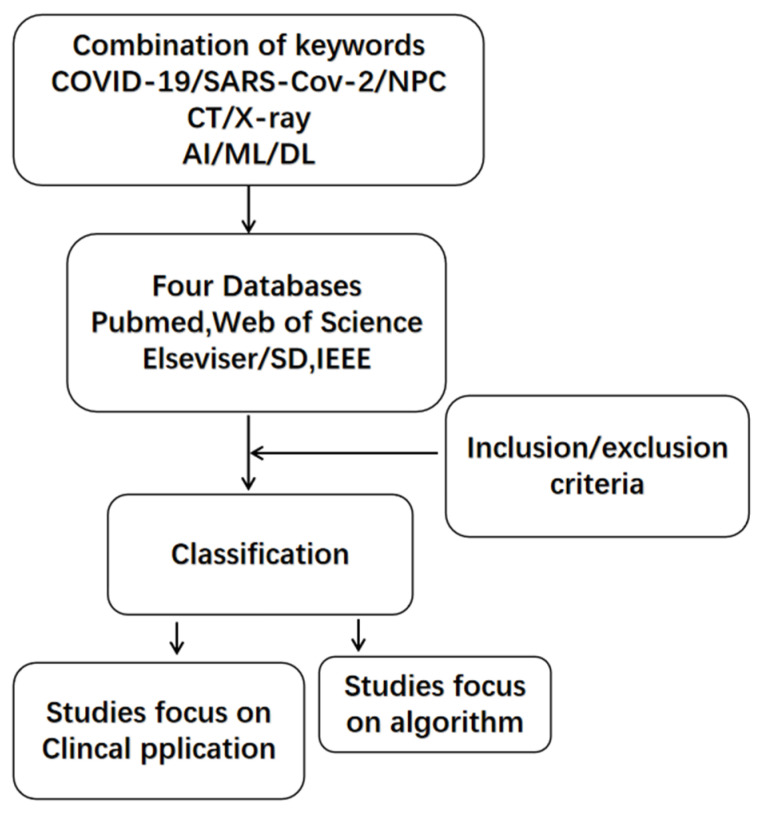
Flowchart of selection and exclusion.

**Table 1 diagnostics-11-01924-t001:** Partial DL and AI research based on COVID-19 radiology images.

First Author	Modality	Technique	Subject	Numbers of Cases	Nation	Sensitivity (%)	Specificity (%)	Accuracy (%)/AUC
Zhang [9]	CT	uAI	Huoshenshan Hospital	2460 patients	China	N/A	N/A	N/N
Li [10]	CT	COVNet	6 medical centers	4352 scans/3322 patients	China	90	96	N/0.96
Bai [28]	CT	EfficientNet B4 DNN	10 medical centers	1186 patients/132,583 slices)	multinational	95	96	96/0.95
Harmoon [13]	CT	AH-Net/Densnet-121	multinational datasets	2724 scans/2617 patients	multinational	N/A	N/A	N/0.949
Wang [17]	CXR	DeepLabv3	CC-CXRI/CC-CXRI-P	SYSU 145202 CC-CXRI-P 16196	China	92.94	8704	N/0.968
Wehbe [18]	CXR	DeepCOVID-XR	multicenter	2214 images/1192 COVID-19	US	75	93	83/0.90
Jiao [44]	CXR	U-Net	2 medical centers	1834 patients	US	73.8	85.3	N/0.846
Yu [34]	CT	CNN	multicenter	421 patients	China	N/A	N/A	N/N
Mei [41]	CT	CNN	18 medical centers	905 patients	China	N/A	N/A	N/0.92
Hwang [27]	CXR	Lunit INSIGHT CXR 2	4 medical centers	172 CXRs/172 patients	Korea	71.3	52.2	0.714
Fung [25]	CT	3D ResNet /LSTM	2 medical centers	1040 mild type/2543 patients	China	N/A	N/A	0.92
Wang [23]	CT	Full-uAI-Discover-NCP	single medical center	31 patients	China	N/A	N/A	N/N
Shan [29]	CT	VB-Net	single medical center	549 patients	China	N/A	N/A	N/N
Murphy [26]	CXR	CAD4COVID-XRay	3 medical centers	24,678 patients	the Netherlands	85	61	N/0.81
Li [10]	CXR	Siamese neural network	CheXpert/1 medical center	468 patients	US	N/A	N/A	N/0.80
Lessman [31]	CT	CORADS-AI	1 academic center/1 hospital	843 patients	the Netherlands	85.7	89.8	N/0.95
Wang [21]	CT	DenseNet121-FPN/COVID-19Net	multicenter	5372 patients	China	78.93	89.93	N/0.90
Schiaffino [43]	CT	SVM vs. MLP	6 medical centers	897 patients	Italy	N/A	N/A	N/0.747 vs. 0.844
Bartolucci [33]	CT	3DSlicer/RadAR	2 medical centers	115 patients	Italy	N/A	N/A	N/0.82
Purkayastha [42]	CT	CNN	multicenter	981 patients	multinational	N/A	N/A	N/0.868
Ma [24]	CT	Advanced decision tree based machine	single medical center	244 patients	China	82	84	N/0.84
Cai [20]	CT	RF	single medical center	99 patients	China	N/A	N/A	N/0.917 –0.940
Yue [22]	CT	logistic regression /random forest		31 patients/ 72scans	China	100/75	100/89	N/ 0.97/0.92
Shiri [45]	CT	ResNet	9 medical centers	1141 patients	Switzerland	N/A	N/A	N/N
Sengupta [46]	CT	QNN	5 open-source datasets	9500 + patients	India	97.7	N/A	96.92/ N
Wu [47]	CT	COVID-AL	CC-CCII	962 patients	China	N/A	N/A	86.6/0.968
Fouladi [51]	CT	ResNet-50/VGG-16/CNN/CAENN	COVID-19 dataset	2482 patients	Iran	N/A	N/A	94/N

N or N/A: not applicable.

**Table 2 diagnostics-11-01924-t002:** Partial available open-source datasets of COVID-19 radiology images *.

No.	Source	Contents and Number of Images	Types of Images	Image Format	Links (10 September 2021)
1	COVID-19 Radiography Database	COVID-19 images: 3616, normal images: 10,192 viral pneumonia images: 1345	CXR	PNG	https://www.kaggle.com/tawsifurrahman/covid19-radiographydatabase
2	COVID-19 Dataset	COVID-19 images: 3616	CXR	DCM	
3	J. P. Cohen’s GitHub	N/A	CXR	JPG and PNG	https://github.com/ieee8023/covid-chestxray-dataset
4	CC-CCII	Total 617,775 COVID-19, CP and normal	CT	JPG	http://ncov-ai.big.ac.cn/download
5	UCSD-AI4H	COVID-19 images: 349, non-COVID-19 images: 397	CT	JPG and PNG	https://github.com/UCSD-AI4H/COVID-CT
6	HUST-19-iCTCF	Total 19,685	CT	DCM and JPEG	http://ictcf.biocuckoo.cn/
7	European Society of Radiology	N/A	CXR and CT	PDF	https://www.eurorad.org/advanced-search?search=COVID
8	COVID-19-AR	CXR 233 and CT23 31,935 images/105 patients	CXR and CT	DCM	https://wiki.cancerimagingarchive.net/pages/viewpage.action?pageId=70226443
9	COVIDx Dataset	CXR 16,352 and CT 194,922	CXR and CT	DCM	https://github.com/lindawangg/COVID-Net

* Some datasets are still expanding images, so there may be a delay in the data presented in the table. N/A or N: not applicable.

## Data Availability

No new data were created or analyzed in this study. Data sharing is not applicable to this article.

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
