# Peer review of "A Promising and Challenging Approach: Radiologists’ Perspective on Deep Learning and Artificial Intelligence for Fighting COVID-19"

_diagnostics, 2021, doi:10.3390/diagnostics11101924_

Round 1
Reviewer 1 Report
Interesting study on AI and machine learning to explore COVID19.
The first person must be avoided (our, we).
Verbiage and broad statement must be deleted: see first sentence abstract.
The methodology of this review is missing.
Flowchart of reference selection/exclusion may be valuable.
The manuscript has to be shorten and the authors should refer to the tables instead of detailed each single study.
Each statement must be supported by references.
Please define all acronyms, see DL vs ML?
The rationale and aims of this review must be clearly stated as the reader is lost under all the informations provided.
The authors should summarize the findings of this study with some key points easily understandable by readers.
Author Response
Reviewer #1:
Comments and Suggestions for Authors
Interesting study on AI and machine learning to explore COVID19.
Question 1 .The first person must be avoided (our, we).
Response: Thanks for your suggestions, we revised and checked the manuscript again to make sure there is no first person anymore.
Question 2.Verbiage and broad statement must be deleted: see first sentence abstract.
Response: Thanks for your suggestions. Mojor revisions have been made in the part of Abstract to reflect more actual content and results of the manuscript. In addition, we also delete broad statements in other parts of the article.
Question 3.The methodology of this review is missing.
Response: Thanks for the viewer's comments.This manuscript is a rapid review.As far as we know, the methodology of the review method of rapid review is optional,but not necessary, which is not like a scope review or systematic review or meta-analysis strictly required by guidelines like PRISMA and EQUATOR. Explaining review method would not increase the readability and comprehension of this quick review.we would love to show our methodology to the reviewers.
A combination of keywords including "COVID 19", "SARS-CoV-2", "NPC", "CT", "X-ray" and "machine learning", "deep learning", and "artificial intelligence" was used to searched for articles in the four databases including PubMed, Web of Scicence , Elsevier/SD and IEEE, and all types of research were evaluated, including case report, original research, clinical guidelines and systematic reviews. Through reading abstracts and reviewing the references of included studies, the published studies are finally screened and classified. We set the criteria for inclusion and classification. Only research written in English is included, preprints that have not been peer-reviewed are excluded. And as the title points out, The application and effectiveness of these systems were discussed from the perspective of radiologists. The types and developments of algorithms were not the focus of this review, but they were not excluded.
Question 4 .Flowchart of reference selection/exclusion may be valuable.
Response: Thanks for the viewer's suggestions.we added an flowchart of reference selection/exclusion.
Question 5 .The manuscript has to be shorten and the authors should refer to the tables instead of detailed each single study.
Response: Thanks for the comments. we have already shown parts of results of some reference in the Table 1. The reason for choosing a more detailed introduction is to fully show the differences in different studies which causing some redundancy actually , but another reviewer's opinion thinks that these studies should introduce more detail of especially about figures. To be honest, this situation makes us In a dilemma. So we made appropriate changes in the revised manuscript based on the comments of the reviewers, mainly highlighting the difference of performance between the external test and the internal test in the form of numbers figures.
Question 6 .Each statement must be supported by references.
Response: Thanks to the comments, we have read the manuscript carefully, and added references where need citation support.
Question 7 .Please define all acronyms, see DL vs ML?
Response: Thanks for your suggestions,We revised the manuscript and defined acronyms. In addition, we added some knowledge about machine learning and deep learning in the introduction part and added a reference, and added Figure 1 to increase readers’ understanding.
Question 8 The rationale and aims of this review must be clearly stated as the reader is lost under all the informations provided.
Response:Thanks to your comments.we have added the basic principles and aims of this rapid review in the part Introduction to increase readers’ understanding.
Question 9 The authors should summarize the findings of this study with some key points easily understandable by readers.
Response: Thanks for your suggestions.We summarized this study at the part Abstract, Introduction, and the end of article, and performaned introdution and summary in each part. In addition, the results and significance of different research were summarized in the manuscript.
Reviewer 2 Report
In this review the authors have analyzed the current status of artificial intelligence in the elective field of COVID-19.
This is a well written and comprehensive review.
The should authors should make the following changes
1. The Abstract should be more descriptive with actual results.
- In general, the authors should use “correlation” or “correlated” when cited papers actually did a correlation study. If not, the word “associated” should be preferred
- A reference dealing with machine learning and deep learning should be provided in the Introduction. Please use the following one
Nakaura T, Higaki T, Awai K, Ikeda O, Yamashita Y. A primer for understanding radiology articles about machine learning and deep learning. Diagn Interv Imaging 2020;101(12):765-770.
- In the limitation section, the authors should add that Ai is not the the ultimate solution right now and that simple models exist to determine the severity of the disease and add the following reference
Devie A, Kanagaratnam L, Perotin JM, Jolly D, Ravey JN, Djelouah M, Hoeffel C. COVID-19: A qualitative chest CT model to identify severe form of the disease. Diagn Interv Imaging 2021;102(2):77-84.
- In the conclusion, the authors should stress that currently most of COVID-19 imaging is made without artificial intelligence and that the human eye is still needed and add the following reference
Li J, Long X, Wang X, Fang F, Lv X, Zhang D, Sun Y, Hu S, Lin Z, Xiong N. Radiology indispensable for tracking COVID-19. Diagn Interv Imaging 2021;102(2):69-75.
- The most recent references about AI and COVID-19 should be added
Wang R, Jiao Z, Yang L, Choi JW, Xiong Z, Halsey K, Tran TML, Pan I, Collins SA, Feng X, Wu J, Chang K, Shi LB, Yang S, Yu QZ, Liu J, Fu FX, Jiang XL, Wang DC, Zhu LP, Yi XP, Healey TT, Zeng QH, Liu T, Hu PF, Huang RY, Li YH, Sebro RA, Zhang PJL, Wang J, Atalay MK, Liao WH, Fan Y, Bai HX. Artificial intelligence for prediction of COVID-19 progression using CT imaging and clinical data. Eur Radiol 2021 ; doi: 10.1007/s00330-021-08049-8.
Keidar D, Yaron D, Goldstein E, Shachar Y, Blass A, Charbinsky L, Aharony I, Lifshitz L, Lumelsky D, Neeman Z, Mizrachi M, Hajouj M, Eizenbach N, Sela E, Weiss CS, Levin P, Benjaminov O, Bachar GN, Tamir S, Rapson Y, Suhami D, Atar E, Dror AA, Bogot NR, Grubstein A, Shabshin N, Elyada YM, Eldar YC. COVID-19 classification of X-ray images using deep neural networks. Eur Radiol 2021 ; doi: 10.1007/s00330-021-08050-1.
Mortani Barbosa EJ Jr, Georgescu B, Chaganti S, Aleman GB, Cabrero JB, Chabin G, Flohr T, Grenier P, Grbic S, Gupta N, Mellot F, Nicolaou S, Re T, Sanelli P, Sauter AW, Yoo Y, Ziebandt V, Comaniciu D. Machine learning automatically detects COVID-19 using chest CTs in a large multicenter cohort. Eur Radiol 2021;. doi: 10.1007/s00330-021-07937-3.
- The authors should pay attention to the presentation of the references as some of them have journal names in full and not as abbreviated journal names
Author Response
Reviewer #2:
Comments and Suggestions for Authors
In this review the authors have analyzed the current status of artificial intelligence in the elective field of COVID-19.
This is a well written and comprehensive review.
The should authors should make the following changes
Question 1.1.The Abstract should be more descriptive with actual results.
Response:Thanks for your comments. A wide range of revisions have been made to the part Abstract to reflect more actual content and results.
Question 2.In general, the authors should use “correlation” or “correlated” when cited papers actually did a correlation study. If not, the word “associated” should be preferred
Response:Thanks for the suggestion. We have checked the three references which used "correlation" or "correlated" again, and it is clear that the authors have carried out a correlation analysis in the article.
Question 3. A reference dealing with machine learning and deep learning should be provided in the Introduction. Please use the following one
Nakaura T, Higaki T, Awai K, Ikeda O, Yamashita Y. A primer for understanding radiology articles about machine learning and deep learning. Diagn Interv Imaging 2020;101(12):765-770.
Response:Thanks for the suggestion.We added some knowledge about machine learning and deep learning and added an available reference, and added picture 1 to increase readers’ understanding in the part Introduction.
Question 4. In the limitation section, the authors should add that AI is not the the ultimate solution right now and that simple models exist to determine the severity of the disease and add the following reference
Devie A, Kanagaratnam L, Perotin JM, Jolly D, Ravey JN, Djelouah M, Hoeffel C. COVID-19: A qualitative chest CT model to identify severe form of the disease. Diagn Interv Imaging 2021;102(2):77-84.
Response:Thanks for the great suggestions, we have made modified it in the part Limitation section and added the available reference.
Question 5. In the conclusion, the authors should stress that currently most of COVID-19 imaging is made without artificial intelligence and that the human eye is still needed and add the following reference
Li J, Long X, Wang X, Fang F, Lv X, Zhang D, Sun Y, Hu S, Lin Z, Xiong N. Radiology indispensable for tracking COVID-19. Diagn Interv Imaging 2021;102(2):69-75.
Response:Thanks for the great suggestions, we have made modified it in the part Limitation section and added the available reference.
Question 6. The most recent references about AI and COVID-19 should be added
Wang R, Jiao Z, Yang L, Choi JW, Xiong Z, Halsey K, Tran TML, Pan I, Collins SA, Feng X, Wu J, Chang K, Shi LB, Yang S, Yu QZ, Liu J, Fu FX, Jiang XL, Wang DC, Zhu LP, Yi XP, Healey TT, Zeng QH, Liu T, Hu PF, Huang RY, Li YH, Sebro RA, Zhang PJL, Wang J, Atalay MK, Liao WH, Fan Y, Bai HX. Artificial intelligence for prediction of COVID-19 progression using CT imaging and clinical data. Eur Radiol 2021 ; doi: 10.1007/s00330-021-08049-8.
Keidar D, Yaron D, Goldstein E, Shachar Y, Blass A, Charbinsky L, Aharony I, Lifshitz L, Lumelsky D, Neeman Z, Mizrachi M, Hajouj M, Eizenbach N, Sela E, Weiss CS, Levin P, Benjaminov O, Bachar GN, Tamir S, Rapson Y, Suhami D, Atar E, Dror AA, Bogot NR, Grubstein A, Shabshin N, Elyada YM, Eldar YC. COVID-19 classification of X-ray images using deep neural networks. Eur Radiol 2021 ; doi: 10.1007/s00330-021-08050-1.
Mortani Barbosa EJ Jr, Georgescu B, Chaganti S, Aleman GB, Cabrero JB, Chabin G, Flohr T, Grenier P, Grbic S, Gupta N, Mellot F, Nicolaou S, Re T, Sanelli P, Sauter AW, Yoo Y, Ziebandt V, Comaniciu D. Machine learning automatically detects COVID-19 using chest CTs in a large multicenter cohort. Eur Radiol 2021;. doi: 10.1007/s00330-021-07937-3.
Thanks for your kindly suggestion.
Response:Thanks for your kindly suggestion. There are still a large number of studies published as we mentioned in the article and the table, and this rapid review can only reach a certain stage at present, and it is impossible to include all studies,especially many studies are similar or recently updated.we have added these available references.
Question 7.The authors should pay attention to the presentation of the references as some of them have journal names in full and not as abbreviated journal names
Response:Thanks for your kindly reminder. As suggested, we have revised the corresponding sentence in the revised manuscript.
Reviewer 3 Report
This review paper provides an overview of the latest research to apply machine learning techniques to aid COVID-19 diagnosis and treatment. The review focusses on the applications, rather than the ML techniques, and provides a reasonable balance between reported successes and current limitations (particularly due to model training and robustness). The ongoing research to combine image data with other medical data in a predictive model is of particular interest. The paper is generally well organised. However some of the results of papers are quoted without providing further insight and so some of the key conclusions or discussion points can become lost among the quoted ROC and AUC values. I also note that there are no figures in the manuscript. Inclusion of figures would demonstrate the complexity of the applications and allow the reader to visually judge the success of the model outputs. Therefore I have a few suggests to improve the review:
- More focus in the text should be given to the higher quality research papers (e.g where successful models have been tested on many images / data sets from different institutes) with some examples of the model outputs given in figures.
- While the ML techniques themselves are not the main focus of the review, more insight should be given as to why certain models have performed better than others, and which techniques look the most promising for the future for different applications. For example you review on line 151 the work of Fung et al who developed a 2-stage ML model - why was such a model an improvement over others in general terms?
- Can the authors provide some discussion on relative success of ML techniques applied to different imaging modalities. For example, most research seems to have been applied to CT data. How does models developed for x-ray radiography generally compare?
- It would be useful to add, perhaps as an additional table, which models are open access and are available to download and try.
Author Response
Reviewer #3:
This review paper provides an overview of the latest research to apply machine learning techniques to aid COVID-19 diagnosis and treatment. The review focusses on the applications, rather than the ML techniques, and provides a reasonable balance between reported successes and current limitations (particularly due to model training and robustness). The ongoing research to combine image data with other medical data in a predictive model is of particular interest. The paper is generally well organised. However some of the results of papers are quoted without providing further insight and so some of the key conclusions or discussion points can become lost among the quoted ROC and AUC values. I also note that there are no figures in the manuscript. Inclusion of figures would demonstrate the complexity of the applications and allow the reader to visually judge the success of the model outputs. Therefore I have a few suggests to improve the review:
Question 1.More focus in the text should be given to the higher quality research papers (e.g where successful models have been tested on many images / data sets from different institutes) with some examples of the model outputs given in figures.
Response:Thanks for your comments. We show which studies used data from multi-institution for testing in Table 1. We found a large number of studies using similar methods when we performaned this rapid review, but the number of high-quality studies is relatively small. For example, we consider the study by Zhang et al.to be a high-quality study, and we have cited it three times. In addition, we added a discussion about the following two questions of the reviewer using their research as an illustration (in Part 10 Limitations and Challenges). We focused on the presence or absence of external verification, and all the research results with external verification or prospective verification have been introduced in the manuscript. We also found that most studies tend to collect data from multiple institutions for testing, but lack of external verification by multiple institutions.
Question 2.While the ML techniques themselves are not the main focus of the review, more insight should be given as to why certain models have performed better than others, and which techniques look the most promising for the future for different applications. For example you review on line 151 the work of Fung et al who developed a 2-stage ML model - why was such a model an improvement over others in general terms?
Response: Thanks for your comments: Our team's internal discussions and discussions with AI experts believe that the performance of the model is affected by several factors: imaging mode, data source, datasets size, algorithm choise, as we listed in the manuscript. We discussed this queestion added in Part 10. At present, the evaluation of all models or algorithms is based on diagnostic performance, which is the result theory. There Is No Best, Only a Better Fit. We have also seen that some models have been improved in performance through improvements to some algorithms, but these improvements are not always effective. As we mentioned in the manuscript, transfer learning and data enhancement are common strategies to solve the scarcity of data. However, when the sample size is sufficient, strategies such as transfer learning will affect the computing efficiency and generalization of the system.We look forward to more advanced models and algorithms provided by engineers and technologists in the future.
Question 3.Can the authors provide some discussion on relative success of ML techniques applied to different imaging modalities. For example, most research seems to have been applied to CT data. How does models developed for x-ray radiography generally compare?
Response: Thanks for your suggestions.Tthe establishment of models for different imaging modes is affected by regions. Research institutions from China mainly and use CT data models for the recommend the use of CT,while the United States and other countries from Europe have more applications of CXR AI,more AI studies of CXR data models have been produced. It is not difficult to conclude by observing Table 1 that the model performance using CT data is usually better than the model using CXR data . We have added corresponding content in the PART 10. The main reason is considered to be the inherent shortcomings of CXR images. The ordinary CXR image is the sum of the effects of X-rays on all tissues between the X-ray source and the capture membrane, , the tissue structure in X-rays is not clear and lack of three-dimensional information compared with CT images.
Question 4.It would be useful to add, perhaps as an additional table, which models are open access and are available to download and try.
Response: Thanks for the suggestion, we have added an additional table of open codes.
Round 2
Reviewer 1 Report
The authors adequately addressed the comments of reviewers.